

# Northward shift of boreal tree cover confirmed by satellite record

**Authors**

Min Feng[1,2]; Joseph O. Sexton[1]; Panshi Wang[1]; Paul M. Montesano [3,4]; Leonardo Calle [5]; Nuno Carvalhais [6,7]; Benjamin Poulter [8,9]; Matthew J. Macander [10]; Michael A. Wulder [11]; Margaret Wooten [3,12]; William Wagner [3,12]; Akiko Elders [13]; Saurabh Channan [1]; Christopher S.R. Neigh [3,12]

**Affiliations**

[1] terraPulse, Inc., North Potomac, Maryland, USA.

[2] Institute of Tibetan Plateau Research, Chinese Academy of Sciences; Beijing, China.

[3] NASA Goddard Spaceflight Center; Greenbelt, Maryland, USA.

[4] ADNET Systems, Inc.; Bethesda, Maryland, USA.

[5] calleEcology, Inc., Missoula, MT, USA.

[6] Max Planck Institute for Biogeochemistry, Jena, Germany.

[7] Departamento de Ciências e Engenharia do Ambiente, DCEA, Faculdade de Ciências e Tecnologia, FCT, Universidade Nova de Lisboa, Caparica, Portugal.

[8.] Spark Climate Solutions, San Francisco, California, USA.

[9.] Department of Geographical Sciences, University of Maryland, College Park, Maryland, USA.

[10] ABR, Inc.—Environmental Research & Services, Fairbanks, Alaska, USA.

[11] Canadian Forest Service (Pacific Forestry Centre), Natural Resources Canada, 506 West Burnside Road, Victoria, British Columbia, Canada.

[12] Science Systems Applications, Inc., Lanham, Maryland, USA.

[13] Morgan State University, Baltimore, Maryland, USA.

Correspondence to: Min Feng (mfeng@terrapulse.com), Joseph O. Sexton (sexton@terrapulse.com)



**Abstract.** The boreal forest has experienced the fastest warming of any forested biome in recent decades. While
vegetation–climate models predict a northward migration of boreal tree cover, the long-term studies required to test
the hypothesis have been confined to regional analyses, general indices of vegetation productivity, and data calibrated
to other ecoregions. Here we report a comprehensive test of the magnitude, direction, and significance of changes in
the distribution of the boreal forest based on the longest and highest-resolution time-series of calibrated satellite maps
of tree cover to date. From 1985 to 2020, boreal tree cover expanded by 0.844 million km², a 12% relative increase
since 1985, and shifted northward by 0.29° mean and 0.43° median latitude. Gains were concentrated between 64°–
68°N and exceeded losses at southern margins, despite stable disturbance rates across most latitudes. Forest age
distributions reveal that young stands (≤36 years) now comprise 15.4% of forest area and hold 1.1–5.9 Pg of
aboveground biomass carbon, with the potential to sequester an additional 2.3–3.8 Pg C if allowed to mature. These
findings confirm the global advance of the boreal forest and implicate the future importance of the region's greening
to the global carbon budget.

**1 Introduction**
The boreal biome is Earth's most expansive and ecologically intact forest. The region contains $38 \pm 3.1$ Pg C of above-
ground biomass (Neigh et al., 2013) and is underlain by 1672 Pg C, summing to total biomass rivaling the tropics and
half of global soil C (Gauthier et al., 2015). Its forested area comprises a third of the global total and accounts for
20.8% of the total forest carbon (C) sink (Pan et al., 2011). Boreal tree cover also controls the reflective and thermal
balance of solar radiation of the high northern latitudes, posing a positive feedback mechanism for greenhouse
atmospheric warming (Betts, 2000; Bonan, 2008; Chen et al., 2018; Randerson et al., 2006).

48        The boreal region has experienced the fastest climatological warming of any forest biome, with annual
surface temperatures increasing $> 1.4°$ C over the past century (IPCC, 2014). Boreal forest dynamics are highly
correlated to climate (Elmendorf et al., 2012; Holtmeier and Broll, 2005; Véga and St-Onge, 2009), and increases in
vegetation productivity have been observed across the northern high latitudes (Berner and Goetz, 2022). However,
regional increases in the frequency and severity of windthrow, fire, insect, and disease events have also been reported
(Gauthier et al., 2015; Walker et al., 2019), and a recent analysis by Rotbarth et al. (2023) suggests that southern
contraction exceeds northern expansion, yielding net shrinkage of the boreal forest.

55        While theory predicts a northward shift of the boreal forest, the global net effects of climate and other factors
on the density and distribution of its tree cover remain untested hypotheses at the spatial and temporal scale of Landsat,
Earth's longest-running record of global, high-resolution satellite imagery. Coupled climate-vegetation models predict
a net-northward migration of boreal vegetation due to warming (IPCC, 2018; Scheffer et al., 2012), supporting the
dominance of growth processes. Multiple studies (Berner and Goetz, 2022; Sulla-Menashe et al., 2018; Zhu et al.,
2016; Piao et al., 2020) have reported vegetation "greening" (e.g., Berner and Goetz, 2022) based on spectral indices
of plant productivity. However, the ecological effects of trees differ from those of graminoids, shrubs, and other
vegetation, and the comparatively low productivity of boreal ecosystems necessitate long-term analyses that have
historically been limited to either regional scales or uncalibrated data (Beck et al., 2011; Brice et al., 2020; Taylor et



al., 2017; Rotbarth et al., 2023). As a result, the net effect of growth and mortality on the global distribution of boreal
tree cover, and the resulting effect on carbon budgets, remain uncertain.
66        Here we report a global test of the magnitude and direction of boreal-forest change from 1985 to 2020, as
observed through historical satellite records of tree cover calibrated to the boreal biome. We calibrated and expanded
a global tree cover dataset (Carroll et al. 2011, Sexton et al., 2013) to 224,026 Landsat images estimating tree cover
and its changes over the global extent of the boreal forest and adjacent tundra at annual, 30-meter resolution over 36
years (Fig. S1)—the most extensive and highest-resolution record of boreal tree cover to date. This pan-boreal time
series was then subjected to trend analysis to estimate and map the historical direction, rate, and significance of change
across the region, and the resulting estimates of forest age were used to infer impacts on the region's carbon budget.

**2 Methods**
**2.1 Historical retrieval of tree cover**
To improve characterization of boreal forest structure, we calibrated the 250-m resolution, 2000 - 2020 MODIS
Vegetation Continuous Fields (VCF) Tree Cover product (MOD44B Collection 6; Carroll et al., 2011) against a
region-wide sample of airborne lidar measurements, stratifying by topographic and bioclimatic covariates (SI §2–4).
This boreal-specific calibration improved characterization of tree-cover gradients across the boreal region (Fig. S7),
increasing accuracy, decreasing uncertainty, and improving the linear correlation of per-pixel fractional tree cover
estimates to reference measurements (Fig. S8). Mean absolute error (MAE) decreased to 11.13%, root-mean-squared
error (RMSE) decreased to 16.44%, and the coefficient of determination ($R^2$) of the linear model between estimated
and measured data increased to 0.60. Following calibration, the calibrated MODIS VCF estimates were downscaled
to 30-meter resolution and extended from 1984 to 2020 following Sexton et al. (2013). The residual bias of the
Landsat-based estimates relative to the lidar reference measurements was slight (~2%, SI).
86        Calibrated MODIS VCF estimates were then downscaled to 30-m resolution and extended to 1984–2020 by
applying a machine learning model (gradient-boosted regression tree) to Landsat surface reflectance imagery from
sensors TM, ETM+, and OLI (Sexton et al., 2013; SI §5–6). A total of 224,026 Landsat scenes across 2,189 WRS-2
tiles was used to reconstruct annual tree cover estimates, composited to minimize cloud, snow, and phenological noise.
For each pixel-year, the median value of valid observations was retained, resulting in a consistent, high-resolution
time series of tree cover estimates (Fig. S5–S7).

**2.2. Tree cover trend analysis**
Calibrated, downscaled, and extended tree cover values were then summarized across the region as annual, boreal-
wide means and medians to calculate changes over the 36-year study span (Fig. 2). The annual mean and median tree
cover were also broken down by latitude to calculate the change rate at each latitudinal degree between 47°N to 70°N
(Fig. S10). Cover estimates for 1984 were excluded from the trend analysis due to the poor spatial coverage in the
first operational year of Landsat 5 (Fig. S2), and pixels with ≤ 30 unobscured annual tree cover observations were
excluded to minimize unbalanced representation caused by the lapses in the availability of Landsat images, mainly in
central and northeast Siberia (Neigh et al., 2013; Sexton et al., 2013).






### 2.3. Detection of forest change and estimation of age


Following the United Nations Framework Convention on Climate Change (UNFCCC, 2002), forest was defined as tree cover exceeding 30% within each 30-m pixel. The probability of a pixel being forested, p(F), was calculated as the integral of the probability density function of tree cover values exceeding this 30% threshold (SI §11). Using the 36-year time series of annual, 30-m resolution estimates of forest probability (p(F)), gains and losses were identified by applying a two-sample z-test in a moving kernel centered on transitions across the 50% threshold of p(F) (Fig. S13).


Pixels with multiple statistically significant transitions during the 1985–2020 period were permitted up to three gain or loss events. Disturbances were classified as "complete" if ≤7 years of data were missing, and "incomplete" otherwise. Incomplete disturbances were concentrated in areas with sparse Landsat acquisitions prior to 1999, before implementation of systematic global imaging by Landsat 7 (Sexton et al., 2013; Potapov et al., 2012). Forest age was estimated for each year and pixel by subtracting the year of the most recent significant forest gain from the year of interest. Pixels were classified as "new" forests if no forest cover or loss had been observed earlier in the time series within a 150-m radius (five Landsat pixels); otherwise, forests were considered "recovering." Accuracy of change detection and age estimation was assessed against a reference sample of 2,404 visually interpreted points distributed across the boreal biome (SI Fig. S14–S15).



### 2.4. Estimation of aboveground biomass


Aboveground biomass carbon (AGB) was estimated as a function of forest stand age using a linear growth model (Cook-Patton et al., 2020; Fig. S16), with intercept ($\mu$ = -35.7, $\sigma$ = 12.6) and slope coefficients ($\mu$ = 23.2, $\sigma$ = 3.2) incorporating parametric uncertainty. Because ages of forests older than the 36-year time-series could not be directly observed, we assumed three scenarios of stand age to bracket carbon stock estimates in these undated stands: 36 years yielding 19.1–58.4 Pg C), 100 years yielding 35.8–80.5 Pg C, and 300 years yielding 42.4–89.2 Pg C. These scenarios define the plausible envelope of legacy biomass in mature forest. However, estimates reflected structural biomass only and did not account for potential effects of changes in soil moisture or variation in respiration rates. To contextualize the biomass sink relative to climate-driven emissions, we also evaluated the trend in regional surface air temperature using two reanalysis products. Both records indicated significant warming over the study period, with trends of 0.038°C yr⁻¹ ($r$ = 0.69, $p < 1 \times 10^{-5}$) and 0.035°C yr⁻¹ ($r$ = 0.73, $p < 1 \times 10^{-6}$) respectively (Fig. S17).



### 3 Results


### 3.1. Distribution of boreal tree cover


Tree cover reaches its highest densities in the southern portion of the boreal biome and decreases progressively northward (Fig. 1). Sparse conifer stands, woodlands, herbaceous vegetation, and unvegetated barrens dominate the transition to Arctic tundra, and tree cover is nearly absent north of 71°N. Due to interspersion of tundra, wetlands, and inland water bodies, the most common local (i.e., 30-meter pixel) tree-cover density across the entire boreal forest and taiga-tundra ecotone is below 5%






From 1985 to 2020, boreal tree cover increased by 0.844 million km², a 4.3 percentage point absolute increase
and a 12% relative increase over its 1985 extent (Fig. 1). Tree cover expanded from 7.153 million km² (41.44% of the
region) in 1985 to 7.997 million km² (46.32%) in 2020, with a linear trend of 0.023 million km² yr$^{-1}$ (0.12% yr$^{-1}$;
percent cover = 0.116 × year – 187.6, $R^2$ = 0.99, $p < 0.001$). Applying the UNFCCC forest definition of 10–30% tree
cover (UNFCCC, 2002; Sexton et al., 2016), the region held between 8.95 and 12.41 million km² of forest in 2000
and between 9.41 and 13.26 million km² in 2020.
The latitudinal distribution of tree cover also shifted northward from 1985 to 2020. The mean latitude of tree
cover increased by 0.29°, from 57.37°N in 1985 to 57.66°N in 2020 (mean latitude = 0.0075 × year + 42.6, $R^2$ = 0.79,
$p < 0.001$). The median latitude increased more rapidly, by 0.43° (median latitude = 0.0124 × year + 32.5, $R^2$ = 0.88,
$p < 0.001$), indicating widespread net expansion across the biome rather than outliers of change at either its northern
or southern extremes.





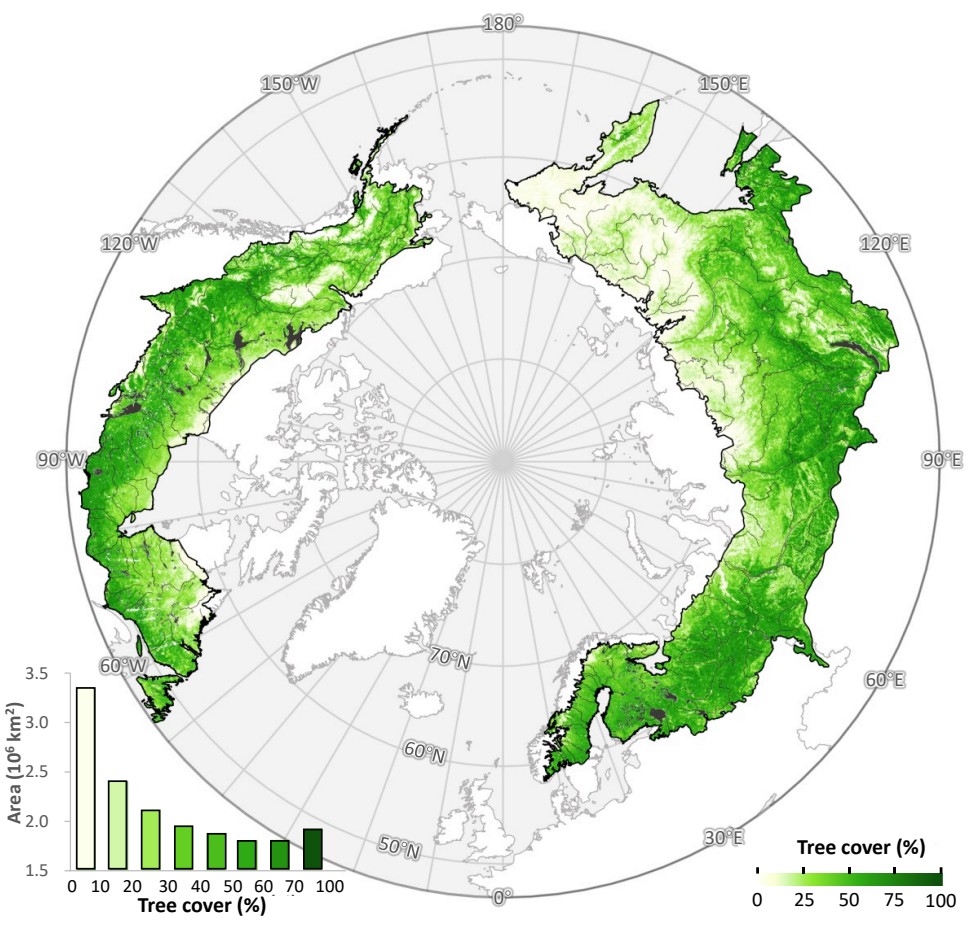

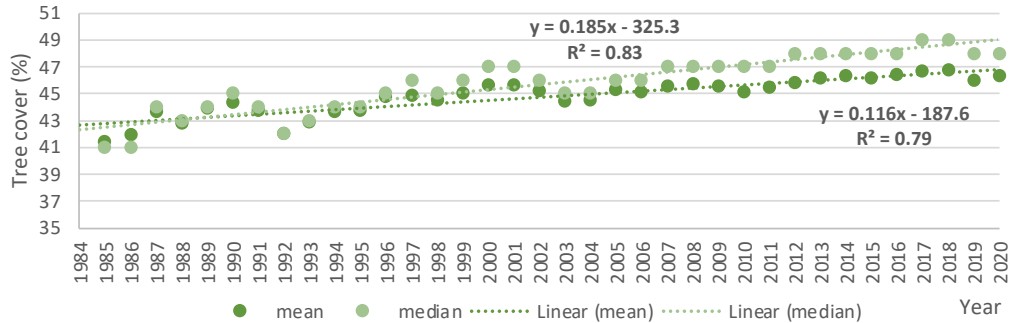


**Fig. 1. Distribution of tree cover across boreal and arctic tundra ecoregions in 2020. Estimates from 2020 are shown. Data**
**gaps due to clouds were filled with estimates from earlier years. Ecoregions were defined by Dinerstein et al (2017). The**
**bottom panel shows the increasing density in the overall, pan-boreal density of tree cover from 1985 to 2020.**





### 3.2. The pace and pattern of boreal forest change


Net biome-wide changes were underlain by strong geographic variation (Fig. 2). Net gains from 1985 to 2020 occurred
at all latitudes above 53°N, with the strongest increases concentrated between 64° and 68°N. Gains in the northernmost
latitudes support the hypothesis of a poleward shift in the northernmost extent of tree cover and are consistent with
findings by Montesano et al. (2024), who reported long-term increases in deciduous and mixed forest components in
transitional boreal zones. These structural shifts parallel recent evidence that warming-induced species diversification
is strongest near the tundra margin as temperate species colonize newly viable habitat (Xi et al., 2024). In contrast,
net losses were smaller in magnitude and limited to the southern boreal latitudes (47°–52°N), corroborating recent
observations by Rotbarth et al. (2023) (Fig. 3).
Our analysis of calibrated, high (30-meter) resolution estimates of tree cover minimized potential for
herbaceous growth to obscure tree mortality, for which coarser-resolution, NDVI-based analyses have been criticized
(Yan et al., 2024). The pan-boreal expansion of tree cover occurred against relatively stable disturbance rates over the
study period (Fig. 3), and observed disturbances influenced regional patterns but did not obscure the biome-wide trend.
The annual rate of disturbance increased modestly from 53,546 km² yr⁻¹ in 2000 to 60,275 km² yr⁻¹ in 2020, equivalent
to a 1.8% yr⁻¹ increase (1,100 km² yr⁻¹), or approximately 0.2%–0.4% of the forested area. Locations undisturbed
between 1985 and 2020 exhibited net gains across nearly all latitudes, and the latitudinal distribution of disturbance—
while varying strongly among years—remained broadly stationary over time. (Fig. S10).
In North America, the largest gains were concentrated in the northernmost boreal, where increases in shrubs
and grasses have also been reported (McManus et al., 2012). Areas of net loss corresponded to widespread forest
disturbances, including wildfire and bark beetle (*Dendroctonus* spp.) outbreaks in British Columbia (Meddens et al.,
2012), spruce budworm (*Choristoneura* spp.) in Quebec (Boulanger and Arseneault, 2004), and wildfire across
western Canada and interior Alaska (Stocks et al., 2002). Recent shifts in transitional forest structure and composition
noted by Montesano et al. (2024) lend further weight to these observations, suggesting a biome-wide response in
functional traits, including increased deciduous dominance at the taiga-tundra ecotone. These findings are also
partially corroborated by Rotbarth et al (2023), who also reported tree cover gains in the boreal interior of North
America but loss at the southern margins, especially in areas impacted by wildfire and harvest.
In Eurasia, hotspots of forest loss included the eastern Russian–Chinese border, agricultural zones south of
the Urals, and regions affected by timber harvesting near the Russia–Finland border in the 1990s (Potapov et al., 2012).
Logging and fire contributed to localized loss in eastern Russia (Krylov et al., 2014), whereas gains in northern Europe
were associated with silvicultural management, afforestation, and fire suppression (Henttonen et al., 2017). Recent
analyses confirm extensive regrowth in post-agricultural and permafrost-transitioning landscapes in Russia, where
lidar and optical remote sensing reveal increases in regeneration potential, particularly in abandoned or disturbed sites
(Neigh et al., 2025).
In Asia, net gains were observed in areas of post-Soviet agricultural abandonment, as well as in larch forests
near the Yakutsk permafrost zone. These trends are consistent with increases in tall shrubs and larch (Larix spp.) at
the taiga–tundra boundary (Frost and Epstein, 2014). Recovery from wildfires in the 1990s continues in these regions
(Kajii et al., 2002), and permafrost thaw has been hypothesized to enhance productivity (Sato et al., 2016).





Although we did not attempt to demarcate or detect changes in a discrete tree line, our observations
corroborate the boreal advancement hypothesis alongside field measurements of woody vegetation near the northern
limits of tree growth and satellite-based studies demarcating the northern tree line (Frost and Epstein, 2014; Rees et
al., 2020; Dial et al., 2024; Dial et al., 2022; Rotbarth et al. 2023). While analysis of tree-cover estimates avoided the
potential confusion of changes in trees specifically with general NDVI-based "greening" (Yan et al. 2024), the trend's
geographic variations correspond to general patterns of greening across the biome (Berner and Goetz, 2022; Sulla-
Menashe et al., 2018; Zhu et al., 2016; Piao et al., 2020; Guay et al., 2014).
Field studies have shown that climate, soil properties, and forest management drive large differences in boreal
tree growth rates across the ecotone (Henttonen et al., 2017; Henttonen et al., 2017; Hofgaard et al., 2009). Recent
shifts in transitional forest structure and composition noted by Montesano et al. (2024) lend further weight to these
observations, suggesting a total biome-wide response in functional traits, including increased deciduous dominance
near treeline margins. Xi et al. (2024) further demonstrate that increasing diversity near the forest–tundra boundary is
associated with moderate climatic warming, although they caution that the gains are vulnerable to reversal under
extremes such as drought and heatwaves. Changes in species composition remain a focal point of research (Xi et al.,
2024; Mekonnen et al., 2019; Massey et al., 2023; Mack et al., 2021; Liski et al., 2003), while still remaining to be
explored are the differentiation of climate and soil effects at the global scale and the discrimination of tree cover
expansion due to the establishment and growth of new seedlings versus the widening of existing tree crowns.





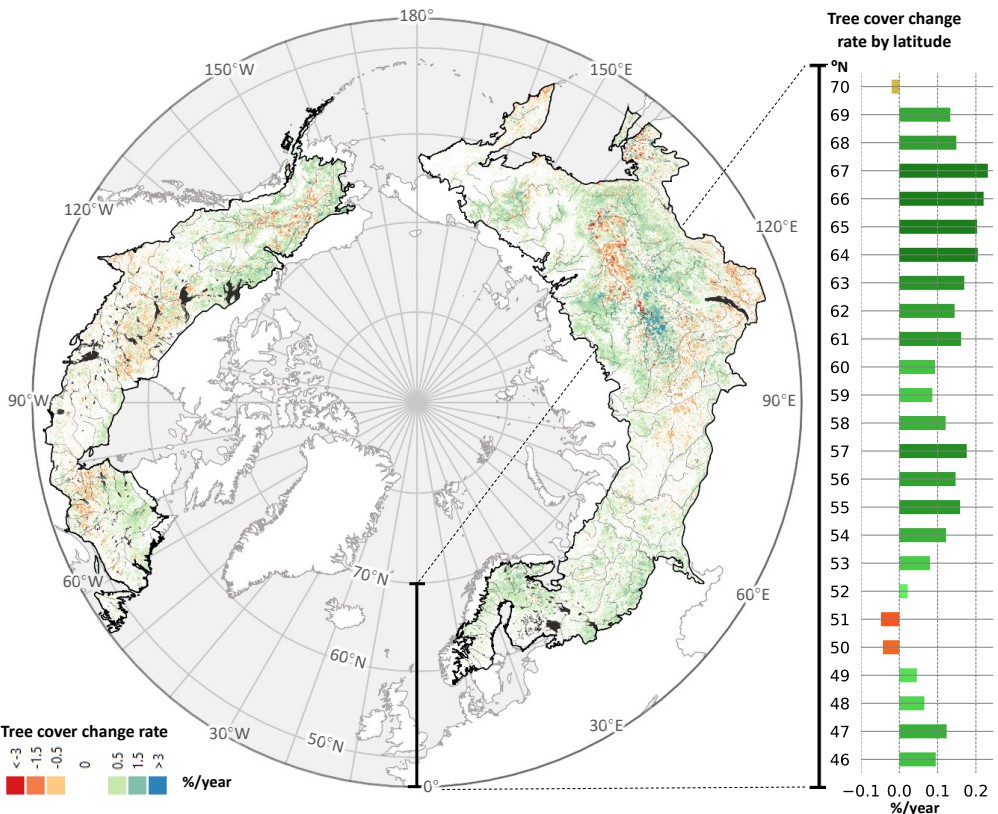


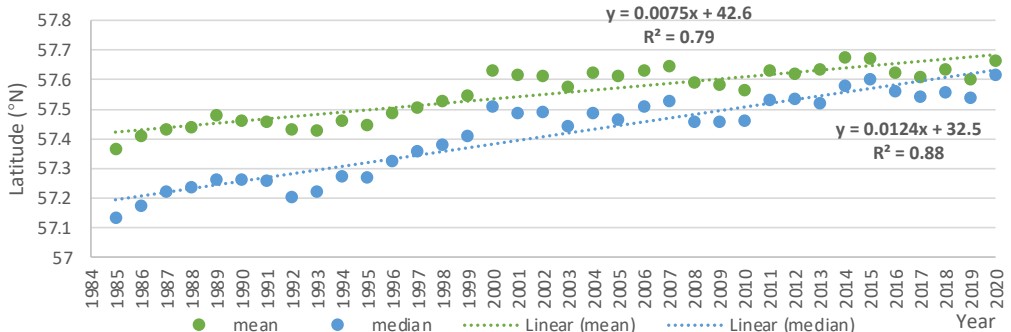

**Fig. 2. Spatial and temporal distribution of boreal tree cover change from 1985 to 2020. Map: significant net gains (green-**
**blue) and losses (orange-red) of tree cover over the boreal biome. Bar chart (top-right): linear regression slope of tree cover**
**over time, stratified by latitude. Time series (bottom): northward migration of the distribution of mean and median latitude**
**of tree cover. Every 30-m resolution pixel included in the analysis had >30 unobscured annual tree cover estimates between**
**1985 and 2020.**



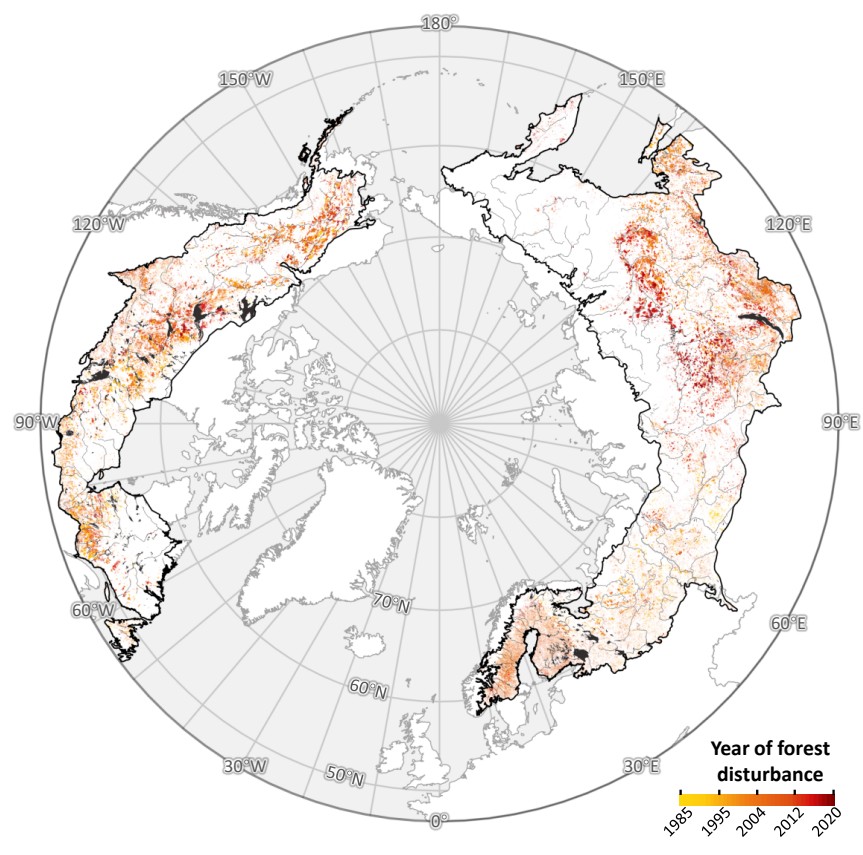


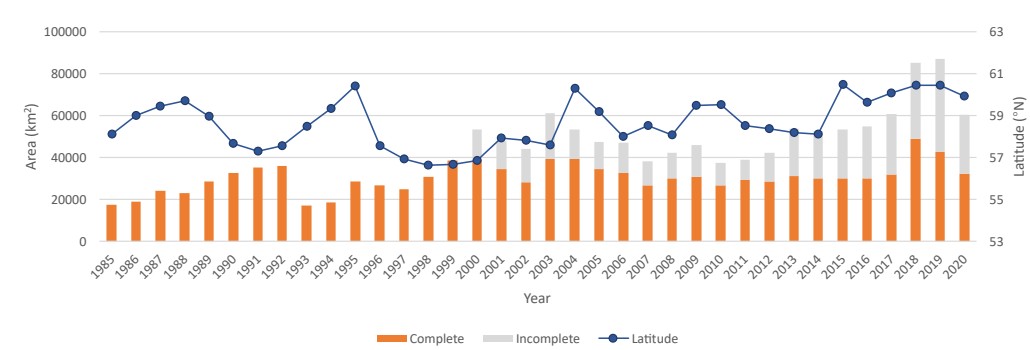


**Fig. 3. Total area and median latitude of boreal stand-clearing disturbances from 1985 to 2020. Trends are plotted for the portion of the boreal area where the satellite image is complete from 1984 to 2019 ("complete") and from all locations, including where the satellite record is incomplete ("incomplete") (Supplemental Information).**





### 3.3. The distribution of boreal forest age

Most of the boreal forest—8.19 million km², or 47.5% of the region—is older than can be directly measured from the satellite record (Fig. 4). Tree cover in these older stands was already established by the beginning of the Landsat observation period in 1985, and the slow rates of biomass accumulation in boreal ecosystems further complicate the detection of recent forest establishment (SI Fig. S15). However, the age of younger stands can be estimated by subtracting the year of first detected forest cover from 2020.

Of the forested area present in 1985, 0.5 million km²—representing 5.29% of standing forests—was disturbed during the study period and recovered to forest by 2020. Recovering forests, combined with "new" forests gained during the Landsat era, produced a weak modal age class centered between 9 and 21 years, with a notable lapse in the youngest age classes. These young forests were concentrated in regions of intensive silviculture, including industrial plantations in Scandinavia (Henttonen et al., 2017; Liski et al., 2003; Ågren et al., 2008), and in areas recovering from wildfire. The latter trend is corroborated by reports of increasing burn frequency and extent in Siberia since the late 20th century (Kharuk et al., 2021), which has driven a rising proportion of recovering forest younger than 20 years.



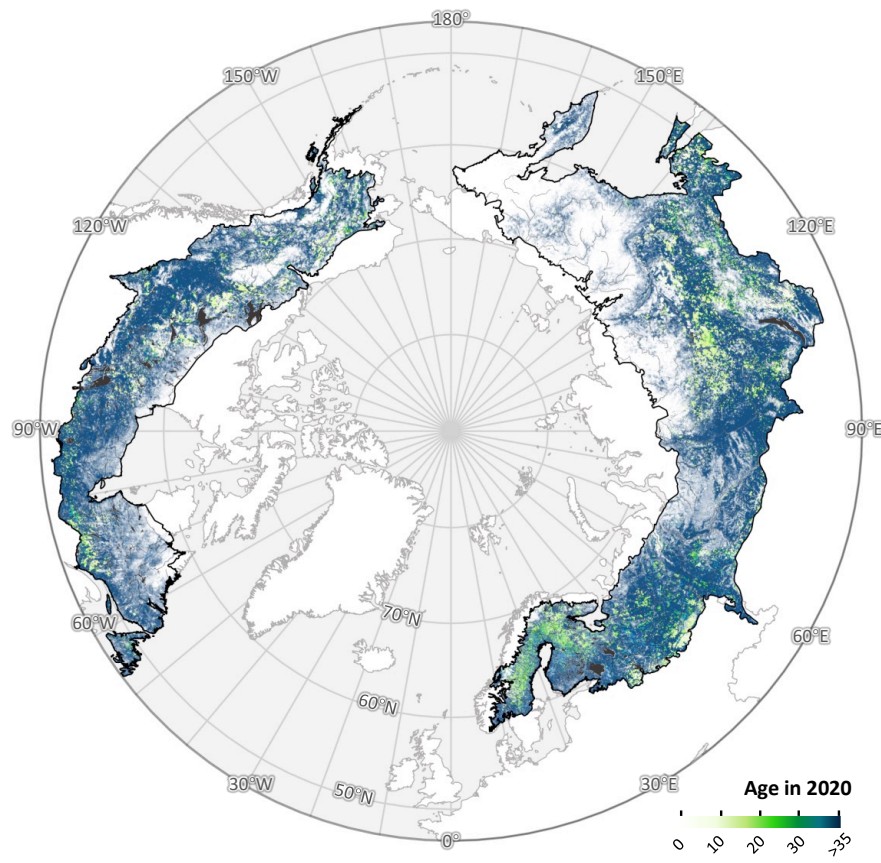

238

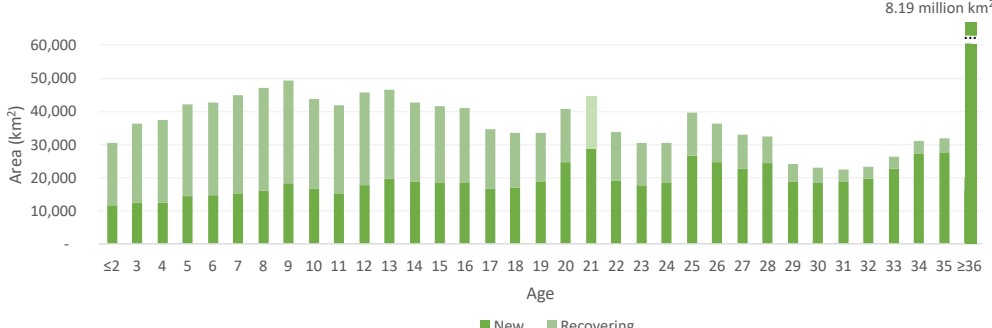

239

**Fig. 4. Spatial distribution of stand age (top) across the boreal ecoregion and frequency distribution of boreal stand age in 2020 (bottom). Forest age-class distribution is defined as years since establishment of pixels identified as forest in 2020. "New" forests were identified as pixels with forest cover following a gain but no prior forest cover or loss earlier in the time series within a 150-m radius (5 pixels) over the observable period (1984 – 2020); "recovering" forests were identified as pixels with forest cover following a gain where a forest loss had been observed previously in the series (Supplemental Information).**

246



## 4 Discussion

The expansion and redistribution of boreal tree cover documented in this study has direct implications for the region's role in the global carbon cycle. Between 1985 and 2020, boreal tree cover increased by 0.844 million km² and shifted northward by over 0.4° in median latitude, with gains concentrated at the biome's northern margin and net expansion observed across most latitudes. These changes are not only spatially extensive but demographically consequential: they reflect a growing fraction of young forests with distinct structural and functional attributes that position them as dynamic agents of carbon sequestration. Understanding the contribution of these forests to current and future carbon stocks is essential for anticipating the net climate feedbacks emerging from boreal ecosystems.

Recent models relating forest age to biomass dynamics suggest that shifting age structure will substantially influence the boreal region's contribution to the global carbon budget in the coming decades. Young forests already contribute significantly to the region's carbon sink (Pan et al., 2011). Forests with known stand ages (≤36 years since disturbance) hold between 1.1 and 5.9 Pg C in aboveground biomass, based on global growth models (Cook-Patton et al., 2020). The ages of forests where no disturbance was observed during the satellite era remain unknown, but plausible aboveground carbon stocks in these older stands can be bracketed between a low-end scenario assuming 36 years of age (19.1–58.4 Pg C) and a high-end scenario assuming 300 years (42.4–89.2 Pg C). Based on these estimates, forests ≤36 years of age comprise 1.35–14.20% of the total boreal aboveground biomass carbon stock—consistent with their 15.4% share of total forest area. Including belowground biomass would raise these values by approximately 25%, based on a mean global root:shoot ratio of 0.25 (Huang et al., 2021).

If allowed to mature without further disturbance, these young forests could sequester an additional 2.3–3.8 Pg C in aboveground biomass. Forests newly established during the observation period contribute between 0.8 and 3.5 Pg C today, exceeding the 0.3–2.4 Pg C held in forests recovering from recorded disturbances. Over the next 36 years, new forests represent a potential additional aboveground sink of 1.3–2.0 Pg C (0.036–0.18 Pg C yr⁻¹), compared to 1.0–1.8 Pg C (0.028–0.05 Pg C yr⁻¹) from recovering forests. This distinction reflects both the greater area occupied by new forests (7.6% vs. 6.7%) and their older mean stand age. These findings support recent observations by Neigh et al. (2025), who reported a disproportionately large contribution of young, regrowing stands to carbon storage in the Russian boreal.

The additional carbon in new forests could help offset warming-induced increases in boreal ecosystem respiration, which have been estimated between 5 and 28 Pg C from 1985 to 2020 (SI Fig. S16). Both climate warming and CO₂ fertilization are expected to enhance productivity (Norby and Zak, 2011), and the spatial pattern of observed tree-cover growth aligns with model predictions of increased seasonal CO₂ exchange above 40°N (Forkel et al., 2016). However, several mechanisms may limit this offset. First, temperature sensitivity of respiration can itself be temperature-dependent (Koven et al., 2017). Second, carbon accumulation rates decline with forest age (Odum, 1969). Third, thawing of permafrost can release substantial legacy carbon stocks (Schuur et al., 2015). Fourth, increases in fire and harvest activity may reverse regional gains in biomass (Gauthier et al., 2015; Kharuk et al., 2021). Compositional and functional transitions may also alter sink dynamics (Montesano et al., 2024; Xi et al., 2024).

The long-term persistence of tree-cover expansion depends not only on productivity, but also on the capacity of boreal soils to support woody vegetation. It remains uncertain whether boreal soils—especially under changing



permafrost regimes—can structurally sustain expanded forest cover (Koven, 2013). Additional uncertainty stems from
the rising role of anthropogenic fire in some parts of the boreal zone (Doerr and Santín, 2016; Mollicone et al., 2006).
Our biomass estimates are derived from models for natural forests and do not account for differences between managed
and unmanaged systems (Kuuluvainen and Gauthier, 2018) or for anticipated changes in fire regimes.

288       While expansion of tree cover may imply increased carbon storage, nonlinear biodiversity responses to
warming complicate projections. Enhanced taxonomic and functional diversity may improve ecological resilience (Xi
et al., 2024), but these benefits are constrained by the growing frequency of climatic extremes. Moreover, biodiversity-
related feedbacks on carbon balance remain difficult to predict under scenarios of increasing disturbance. Ultimately,
all of these processes—forest growth, mortality, disturbance, and compositional change—are already underway across
the boreal biome. Quantifying the balance of autotrophic and heterotrophic carbon fluxes remains critical to
understanding and managing the global climate system.

**Conclusions**

This pan-boreal assessment provides the strongest empirical confirmation to date of a northward shift in boreal tree
cover, long hypothesized by climate–vegetation models. By retrieving the longest, highest-resolution, and most
spatially complete record of calibrated boreal tree cover available, we applied machine learning to the Landsat archive
to reconstruct annual, 30-m maps of forest change from 1985 to 2020. Time-series analysis of $1.9 \times 10^8$ pixels revealed
widespread increases in tree-cover density and a poleward shift in forest distribution, occurring despite relatively
stable disturbance rates across the biome.

302       Although the net trends are globally significant, they mask substantial geographic and temporal heterogeneity,
as well as complexity in the ecological processes underlying forest change. These results underscore the need for high-
resolution, disturbance-aware metrics to supplement NDVI-based assessments, particularly in climatically sensitive
boreal transition zones (Yan et al., 2024). A more complete understanding of boreal forest dynamics will require
integration of satellite time series with field-based measurements of canopy structure and the environmental drivers
of growth, mortality, and species turnover. Moreover, translating the resulting information into action to forestall and
adapt to climate change will require effective communication across scientific, government, and commercial domains
of human activity.

**Acknowledgments**

This research was supported by the NASA Carbon Cycle Science Program (NNH16ZDA001N-CARBON), National
Science Foundation Arctic System Science Program (1604105), and NASA ABoVE (80NSSC19M0112). Satellite
image processing was performed by terraPulse, Inc. on Amazon Web Services (AWS). Reference data for calibration
and validation was produced on the NASA Goddard Spaceflight Center ADAPT and HEC clusters. Aaron Wells (ABR,
Inc.), Celio De Sousa (NASA Goddard Space Flight Center, URSA, Inc.), and Jaime Nickeson (NASA Goddard Space
Flight Center, SSAI, Inc.) contributed reference observations of forest cover and disturbance. Resources supporting



this work were provided by the NASA High-End Computing Program through the NASA Center for Climate
Simulation at Goddard Space Flight Center.
**Author Contributions**
MF and JS designed and developed the tree-cover and forest-change algorithms. PM, PW, and MM conducted the
validation and calibration. CN and PM co-edited the manuscript, CN secured research funding to conduct the
study. BP commented on the final manuscript. NC and LC conducted the carbon impact analysis. NC conducted the
ecosystem respiration analysis. SC developed the platform on AWS. MW, WW, and AE interpreted the validation
dataset. JS conceived the study and compiled the manuscript with contributions from all coauthors.

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
