# Peer review of "Northward shift of boreal tree cover confirmed by satellite record"

_EGUsphere, 2025_

## Referee Comment (RC1)

**Comments on "Northward shift of boreal tree cover confirmed by satellite record" by Min Feng**
**https://doi.org/10.5194/egusphere-2025-2268**

The submitted manuscript reports on the magnitude, direction, and significant changes in the distribution of the boreal forest between latitudes 47°N and 70°N. The analysis is based on estimated MODIS VCF, which were calibrated with lidar observations. These calibrated estimates were then downscaled to 30-m spatial resolution using Landsat data from 1985 to 2020, employing a gradient-boosted regression tree machine learning model.

The authors identified a systematic extension of the boreal forest cover further towards the North pole. This offsets the decrease in tree cover in the southern part of the boreal zone, resulting in an overall increase in boreal tree cover. From a local perspective, different trends in the forest cover patterns and distributions are observed, and further detailed investigation is required in the future. Based on 36 years of satellite data, an estimate of potential $CO_2$ uptake is provided. The authors suggest that this additional uptake could partially compensate anthropogenic $CO_2$ emissions. The authors clearly point out the uncertainties associated with the limited data set and the method applied in the study. Uncertainties arise from, for example, uncertainties in estimated tree age, the influence of temperature on tree growth, the soil's capacity to maintain a growing forest, and anthropogenic fire events. Therefore, the text raises awareness of the potential limitations associated in the provided estimates. Throughout the text, the authors contextualize and compare their findings with recent literature.

I have only some minor comments and I would like to ask the authors to consider them in the next version of the manuscript.

Minor comments:

- L36, L49, and others: Using "≤" in this way seems strange because, strictly following the mathematical interpretation, "increasing" cannot be greater than or equal to "1.4°C." It reads much better written out, and it doesn't add much text. Please consider rewording "<", ">", or "≤" in all instances.

- L37: Please define "C" at the first occurrence. I know that you are referring to carbon, but it should be defined here. It is defined later in line 45, but this is too late.

- L78 and elsewhere: When you cite the supplement, why do you write "SI". What does SI stand for?
  Further, using, e.g., §2-4, does not really help, when the paragraphs in the supplement are not numbered. Having to count the paragraphs from the beginning is cumbersome. Please number the paragraphs in the supplement.

- L83 to L88: Please check: The process of downscaling and calibration of MODIS CVF is mentioned twice! Please remove one of the two instances.

- L88: Even though these abbreviations and acronyms are known in the community, it would be good to explain and give the full names for TM, ETM+, OLI, and WRS-2.

- L97: Maybe start the sentence with "Tree cover [...]" instead of "Cover [...]"?

- L98: As mentioned before, please write "pixels with less than 30 [...]", for example.

- L107: Fig. S13 on one line

- L110: It would be helpful if you could explain what you mean by "disturbance". Although one can understand it after reading the subsequent lines and the supplement, but one should be able to understand the paper without reading the supplement first. For example, provide a short definition of what you regard as a disturbance. Reading the text for the first time, I asked myself if disturbance means: no data, decrease or increase in forest cover, or complete loss of tree cover.

- L117: Try to be consistent. Sometimes you use "SI" when you refer to Fig. in the supplement, like "SI Fig.SX", and other times you only use "Fig.SX". "SI" is not necessary required when writing "Fig. SX", since "S" already directs to the supplement. You may want to consider that.

- L124: Should the bracket behind "58.4 Pg C" be removed?

- L128: "...using two reanalysis products. Both records indicated significant warming... " Please be more specific and state which records you used.  Mentioning the data sets in the supplement is not sufficient, in my opinion and they should be mentioned in the text.

- L137 and L138: After reading the these two sentences, I asked myself what the difference is between "boreal tree cover" and "tree cover". What is meant by the general "tree cover"? Are these different regions? Please explain, and accept my apologies if I missed that.

- Fig1: The right hand side "tree cover" color bar is doubling with the color bar on the left hand side. I do not have a strong opinion about that but you may consider removing the color bar on the right hand side.

- L136: Do you mean Fig.2? Please check?

- L168: Is it fair to calculate the rate of change by selecting two years, here 2000 and 2020? By choosing a different combination, a negative trend could be created, e.g., using 2004 and 2020, which would create a negative trend. Did you calculate the rate over all 10 years or using only the two years?

- L225:  Please be more specific and state which region you are referring to.

- L233: Please specify what you mean by "young," for example: "..., with a notable lapse in the youngest age classes with trees younger than 8 years."

- L264: Please check "root:shoot". Is the colon correct?

- L266: Avoid a line break between the number and unit here and elsewhere.

- L275: Consider explaining "$CO_2$" on the first occurrence.

- L295: You might consider calling it a summary instead of a conclusion. To me, the two paragraphs read more like a summary.

- L298: Please clarify what you mean by "Landsat archive." The Landsat archive contains a variety of products.

Comments concerning the supplement:

- Fig. S4: In the lower most part of the figure, the arrows appear out of nowhere, and directly above, one line ends nowhere. Please check that the boxes and arrows are correctly aligned.

- L225: Please verify: "p(F)" appears three times.

- Fig S14: The red "interpreted points" are difficult to read between the black outlines of the countries. You may want to select a brighter color or one with more contrast.

- L309-311: Please no line break there.

- Fig S16: Please provide a legend in the figure to explain the different colors, particularly for the lowermost left panel. Currently, it is only possible to infer indirectly which color represents regrown forest and which represents new forest. Also, homogenize the subtitles. Sometimes "Forest" appears in the title, and sometimes it does not.

- L347: Q10 is not defined. I guess it's the 10th percentile. Please define it at the first occurrence.

---

## Author Response (AR1)

Dear Editor,

Thank you for the opportunity to revise our manuscript. We also wish to extend our gratitude to the reviewers for their thoughtful and constructive feedback.

We have carefully addressed all comments and believe the manuscript is improved as a result. We have uploaded the revised manuscript along with a point-by-point response to all reviewer comments below.

The main changes in this revision include:

- Improving clarity and definitions of terms and acronyms in the manuscript.

- Adding text to the discussion and methods to explicitly state the study's scope to strengthen the discussion of limitations and scope.

- Adding prominent caveats to the Results and Discussion to highlight uncertainty in forest age estimates.

- Incorporating recent, relevant literature suggested by the reviewers.

We believe these changes have substantially strengthened the manuscript and addressed all reviewer concerns. We look forward to your decision.

Sincerely,

Min Feng (On behalf of all co-authors)

**RC1: ['Comment on egusphere-2025-2268'](), Anonymous Referee #1, 27 Jun 2025**

We thank the reviewer for their thoughtful and constructive feedback on our manuscript. The comments are very helpful for improving the clarity and quality of our paper. We have addressed each of the minor comments below and have revised the manuscript accordingly.

**Comment:** L36, L49, and others: Using "≤" in this way seems strange because, strictly following the mathematical interpretation, "increasing" cannot be greater than or equal to "1.4°C." It reads much better written out, and it doesn't add much text. Please consider rewording "<", ">", or "≤" in all instances.

**Response:** Thank you for this excellent suggestion. We agree that spelling out these operators improves readability and clarity. We have revised the manuscript to write out terms like "greater than," "less than," and "less than or equal to" in all instances within the prose.

**Comment:** L37: Please define "C" at the first occurrence. I know that you are referring to carbon, but it should be defined here. It is defined later in line 45, but this is too late.

**Response:** We appreciate you pointing this out. We have now defined "C" as carbon at its first appearance in the text (L37) to ensure clarity for the reader from the outset.

**Comment:** L78 and elsewhere: When you cite the supplement, why do you write "SI". What does SI stand for? Further, using, e.g., §2-4, does not really help, when the paragraphs in the supplement are not numbered. Having to count the paragraphs from the beginning is cumbersome. Please number the paragraphs in the supplement.

**Response:** Thank you for these very helpful suggestions to improve the usability of our supplement. We have now defined "SI" as "Supplemental Information" at its first use. Furthermore, we have numbered the sections within the supplement to allow for direct and easy navigation. All references to the supplement in the main text have been updated to point to these new numbered sections.

**Comment:** L83 to L88: Please check: The process of downscaling and calibration of MODIS CVF is mentioned twice! Please remove one of the two instances.

**Response:** Thank you for your careful reading. We have reviewed this section and agree that the description was redundant. We have now consolidated and streamlined the text in this paragraph to describe the calibration and downscaling process more concisely and sequentially, removing the repetition.

**Comment:** L88: Even though these abbreviations and acronyms are known in the community, it would be good to explain and give the full names for TM, ETM+, OLI, and WRS-2.

**Response:** We agree. We have now defined these acronyms at their first use in the manuscript: Thematic Mapper (TM), Enhanced Thematic Mapper Plus (ETM+), Operational Land Imager (OLI), and World Reference System 2 (WRS-2).

**Comment:** L97: Maybe start the sentence with "Tree cover [...]" instead of "Cover [...]"?

**Response:** Thank you for the suggestion. We have revised t he sentence to begin with "Tree cover" for improved clarity.

**Comment:** L98: As mentioned before, please write "pixels with less than 30 [...]", for example.

**Response:** We agree. Consistent with our response to the first comment, we have rephrased this to "pixels with 30 or fewer unobscured annual tree cover observations" to improve readability.

**Comment:** L107: Fig. S13 on one line

**Response:** Thank you for catching this formatting issue. We will ensure that "Fig. S13" and all similar figure references appear on a single line in the revised manuscript by using non-breaking spaces.

**Comment:** L110: It would be helpful if you could explain what you mean by "disturbance". Although one can understand it after reading the subsequent lines and the supplement, but one should be able to understand the paper without reading the supplement first. For example, provide a short definition of what you regard as a disturbance. Reading the text for the first time, I asked myself if disturbance means: no data, decrease or increase in forest cover, or complete loss of tree cover.

**Response:** This is an excellent point. A clear definition is essential in the main text. We have now added a concise definition where "disturbance" is first introduced. We clarify that in this study, a disturbance refers to a statistically significant, rapid decrease in tree cover that causes a pixel to transition from a 'forest' state (tree cover > 30%) to a 'non-forest' state.

**Comment:** L117: Try to be consistent. Sometimes you use "SI" when you refer to Fig. in the supplement, like "SI Fig.SX", and other times you only use "Fig.SX". "SI" is not necessary required when writing "Fig. SX", since "S" already directs to the supplement. You may want to consider that.

**Response:** Thank you for pointing out this inconsistency. We agree that the "S" prefix is sufficient. We have standardized all supplemental figure references to the "Fig. S_X_" format throughout the manuscript.

**Comment:** L124: Should the bracket behind "58.4 Pg C" be removed?

**Response:** Yes, thank you for spotting that typo. The misplaced parenthesis has been corrected in the revised manuscript.

**Comment:** L128: "...using two reanalysis products. Both records indicated significant warming... " Please be more specific and state which records you used. Mentioning the data sets in the supplement is not sufficient, in my opinion and they should be mentioned in the text.

**Response:** We agree that this information is important for the main text. We have revised the sentence to explicitly name the datasets used: the Climate Research Unit (CRU) dataset and the European Centre for Medium-Range Weather Forecasts (ECMWF) ERA-Interim reanalysis.

**Comment:** L137 and L138: After reading the these two sentences, I asked myself what the difference is between "boreal tree cover" and "tree cover". What is meant by the general "tree cover"? Are these different regions? Please explain, and accept my apologies if I missed that.

**Response:** Thank you for highlighting this potential ambiguity. In this context, both terms refer to tree cover within our defined boreal study area. To prevent any confusion, we have revised the text to use "boreal tree cover" when discussing the aggregated, biome-wide value and "tree cover" when referring to the variable more generally or at the pixel level. We have ensured the context makes this distinction clear.

**Comment:** Fig1: The right hand side "tree cover" color bar is doubling with the color bar on the left hand side. I do not have a strong opinion about that but you may consider removing the color bar on the right hand side.

**Response:** We agree that the second color bar is redundant. We have removed it from Figure 1 to improve the figure's clarity.

**Comment:** L136: Do you mean Fig.2? Please check?

**Response:** Thank you for this careful check. You are correct; the reference should point to the latitudinal data shown in Figure 2, not Figure 3. We have corrected this reference in the revised manuscript.

**Comment:** L168: Is it fair to calculate the rate of change by selecting two years, here 2000 and 2020? By choosing a different combination, a negative trend could be created, e.g., using 2004 and 2020, which would create a negative trend. Did you calculate the rate over all 10 years or using only the two years?

**Response:** Thank you for this important clarifying question. Our wording was ambiguous. The rate of change was calculated using a linear regression fitted to the annual disturbance data for the entire 2000–2020 period, not from two endpoints. We have revised the sentence to state this explicitly to avoid misinterpretation.

**Comment:** L225: Please be more specific and state which region you are referring to.

**Response:** Thank you for the suggestion. We have revised this sentence to be more specific, noting that young forests are concentrated "in regions of intensive silviculture, such as Scandinavia, and in areas recovering from large-scale wildfire, particularly in Siberia and North America."

**Comment:** L233: Please specify what you mean by "young," for example: "..., with a notable lapse in the youngest age classes with trees younger than 8 years."

**Response:** This is a helpful suggestion. We have revised the text to add a specific age range for clarity, stating there is a lapse in the youngest age classes (e.g., ages 1–8 years).

**Comment:** L264: Please check "root:shoot". Is the colon correct?

**Response:** Thank you for the query. The term "root:shoot ratio" is standard ecological terminology, and the colon is the conventional punctuation used to denote the relationship between the two biomass components. We have retained the original phrasing as it is accurate and widely understood in the field.

**Comment:** L266: Avoid a line break between the number and unit here and elsewhere.

**Response:** Thank you for noting this. We will carefully check the final manuscript layout and insert non-breaking spaces where necessary to ensure that numbers and their corresponding units are not separated across lines.

**Comment:** L275: Consider explaining "CO2" on the first occurrence.

**Response:** Thank you. We have now defined $CO_2$ as carbon dioxide at its first appearance in the manuscript.

**Comment:** L295: You might consider calling it a summary instead of a conclusion. To me, the two paragraphs read more like a summary.

**Response:** Thank you for this thoughtful feedback on the section's framing. We agree that it serves primarily to summarize the study's key findings and their significance. We have renamed the section "Summary and Conclusions" to better reflect its content.

**Comment:** L298: Please clarify what you mean by "Landsat archive." The Landsat archive contains a variety of products.

**Response:** Thank you for this point. To be more precise, we have revised this phrase to specify that we used "the Landsat 4, 5, 7, and 8 surface reflectance archive."

**Comment:** Fig. S4: In the lower most part of the figure, the arrows appear out of nowhere, and directly above, one line ends nowhere. Please check that the boxes and arrows are correctly aligned.

**Response:** Thank you for catching these errors in the flowchart. We have corrected Figure S4 to ensure all boxes, arrows, and lines are properly aligned and connected.

**Comment:** L225: Please verify: "p(F)" appears three times.

**Response:** Thank you for spotting this typo. We have corrected the repetition in the revised supplement.

**Comment:** Fig S14: The red "interpreted points" are difficult to read between the black outlines of the countries. You may want to select a brighter color or one with more contrast.

**Response:** We agree. We have changed the color of the sample points in Figure S14 to a brighter, higher-contrast color (bright cyan) to make them more visible.

**Comment:** L309-311: Please no line break there.

**Response:** Thank you. We have corrected the formatting in this section to remove the unintended line breaks.

**Comment:** Fig S16: Please provide a legend in the figure to explain the different colors, particularly for the lowermost left panel. Currently, it is only possible to infer indirectly which color represents regrown forest and which represents new forest. Also, homogenize the subtitles. Sometimes "Forest" appears in the title, and sometimes it does not.

**Response:** Thank you for these excellent suggestions to improve the figure. We have revised Figure S16 to include a clear legend for the bottom-left panel that distinguishes between 'New Forest' and 'Regrowth'. We have also standardized the subtitles across all four panels for consistency.

**Comment:** L347: Q10 is not defined. I guess it's the 10th percentile. Please define it at the first occurrence.

**Response:** Thank you for highlighting this critical ambiguity. $Q_{10}$ is an important, but specialized, term. It is the temperature coefficient, representing the factor by which a process rate (here, ecosystem respiration) increases with a 10°C rise in temperature. We have now added this explicit definition at its first use in the supplement to prevent misinterpretation.

We are grateful to the reviewer for their positive assessment and insightful comments on our manuscript. We appreciate the recognition of our study's contribution and the specific suggestions for improvement, which we have addressed below.

**Response to General Comments**

- **Comment:** The reviewer notes that excluding tundra ecoregions adjacent to the Arctic Ocean is a limitation and that it would be interesting to know whether tree cover advances can be observed there.
- **Response:** We thank the reviewer for this excellent point. Investigating tree cover changes in the high-Arctic tundra is indeed a critical area for future research. Our study's primary objective was to conduct a comprehensive analysis of the core boreal biome and the immediate taiga-tundra ecotone, which we defined using the boreal and select adjacent tundra ecoregions from Dinerstein et al. (2017) . A robust analysis of the high-Arctic would necessitate different methodologies and calibration datasets tailored to the extremely sparse vegetation and unique remote sensing challenges of that environment. We have added a statement to our discussion acknowledging this as a scope limitation and highlighting the need for future studies focused specifically on these northernmost ecoregions.

**Responses to Specific Comments**

- **Comment:** Line 45: Consider citing Pan et al., 2024 (https://doi.org/10.1038/s41586-024-07602-x).
- **Response:** Thank you for bringing this highly relevant and recent publication to our attention. We have now incorporated and cited Pan et al. (2024) in the manuscript to provide the most current context for our work.

- **Comment:** Line 49: Please reference the more recent IPCC Sixth Assessment Report (IPCC, 2023).
- **Response:** Thank you for this suggestion. We have updated our manuscript to reference the IPCC Sixth Assessment Report, ensuring our background information is current.

- **Comment:** Line 113: Forest age is defined as the "year of the most recent significant forest gain," but trees already have age >0 when first detected as forest in remote sensing. This approach may underestimate stand age. Could you clarify, or indicate whether you tested which minimum tree age is recognized as tree cover?
- **Response:** This is a crucial point. We acknowledge that our satellite-based method inherently provides a conservative estimate of the true stand age. The "age" we

measure is the time since a 30-m pixel crossed the 30% tree cover threshold required to be classified as 'forest'. This approach does not capture the initial years of seedling establishment and growth when cover is below this detection threshold. We have added a clarifying statement to the methods section to explicitly state that our "forest age" refers to "time since detectable forest establishment" and that it represents a minimum age for the stand.

- **Comment:** Line 114: The criterion ("no forest cover or loss within a 150-m radius earlier in the time series") may misclassify forests regenerating from large-scale pre-1984 disturbances, potentially overestimating the area of "new" tree cover.
- **Response:** We thank the reviewer for this sharp observation. This is a valid point and an inherent limitation of any analysis based on the satellite record beginning in the mid-1980s. We cannot observe disturbances that occurred prior to our study period. Consequently, some areas we classify as "new" forest may actually be "recovering" from pre-1985 disturbances. We have added a sentence to the methods and discussion to explicitly acknowledge this uncertainty and clarify that our "new forest" category represents areas with no observed forest cover or loss *within the 1985–2020 satellite observation period*.

- **Comment:** Lines 123–124: The three scenarios of stand age ("100 years yielding 35.8–80.5 Pg C") would benefit from references or a clearer rationale.
- **Response:** Thank you for this request for clarification. We selected these scenarios to bracket a plausible range of carbon stocks for mature forests whose ages predate our satellite record. The 36-year scenario represents the absolute minimum possible age. The 100-year and 300-year scenarios were chosen to represent typical ages for mature and old-growth stands, respectively, in boreal ecosystems. We have now revised the text to clarify this rationale and have added supporting citations regarding typical stand ages in boreal forests.

- **Comment:** Line 139 vs. 142: You report boreal tree cover as 7.997 million km$^2$ in 2020 under a 30% threshold, but then 9.41–13.26 million km$^2$ for the 10–30% threshold. Why does the first figure not fall within the second range?
- **Response:** We appreciate the opportunity to clarify this important distinction. The two figures represent different metrics.
    - The **7.997 million km$^2$** figure is the total **area of tree canopy,** calculated by summing the fractional tree cover within every 30-m pixel across the entire study region[5].
    - The **9.41–13.26 million km$^2$** range represents the total **land area classified as 'forest'**, which includes all pixels where tree cover exceeds a defined threshold (10% or 30%)[6].

      For example, two pixels with 50% cover would contribute their full land area to the 'forest area' calculation but only half their area each to the 'canopy

area' calculation. We have revised the text to make the distinction between 'total canopy area' and 'total land area defined as forest' explicit.

- **Comment:** Line 151: In Fig. 1, much of the tundra biome is missing compared to Dinerstein et al. (2017). Tundra extends to the Arctic Ocean coast.
- **Response:** The reviewer is correct. Our study area was designed to cover the boreal biome and its immediate transition zone with the tundra, not the entire pan-Arctic tundra biome[7]. To avoid confusion, we have revised the caption for Figure 1 and the "Study Area" description to clarify that the map displays our specific study area, which includes only a subset of the global tundra ecoregions delineated by Dinerstein et al. (2017) .

- **Comment:** Lines 181–191: Consider citing more recent studies showing changing trends since 2010, e.g. https://www.nature.com/articles/s41561-022-01087-x.
- **Response:** We thank the reviewer for this helpful suggestion. We have reviewed the recommended study and other recent literature and have incorporated these findings into our discussion to provide a more current and nuanced context for our results, particularly regarding disturbance trends in the last decade.

- **Comment:** Supplementary Fig. S1: According to Dinerstein et al. (2017), most tundra ecoregions are omitted (e.g., Scandinavian Montane Birch forest and grasslands). Only ~7 of 48 tundra ecoregions are shown. It may help to clarify that the analysis focuses on the boreal biome and includes only selected tundra ecoregions where tree cover change was observed.
- **Response:** The reviewer's interpretation is correct. Our focus was on the boreal biome and the taiga-tundra ecotone. We have revised the "Study Area" section and the caption for Figure S1 in the supplement to explicitly state that our analysis includes all boreal forest/taiga ecoregions along with a selection of immediately adjacent tundra ecoregions, and that it does not cover the full extent of the pan-Arctic tundra biome.

**CC1: 'Comment on egusphere-2025-2268', Richard Fernandes, 09 Jun 2025**

*The results presented in Figure 1 and 2 rest on the temporal precision and temporal stability of the tree cover change estimates (see https://library.wmo.int/idurl/4/58111 for definition of these quantities).   The Supplementary information Figure S8 panel b compares calibrated and reference tree cover.  This raises three questions that should be addressed:*

*1.  Figure S8b only shows comparisons between model predictions and LIDAR reference data.  Comparisons should also be presented and summarized for the high resolution reference imagery,*

**Reply to comment:**

We thank the reviewer for raising the important issue of the accuracy and precision of our reference datasets. As the reviewer notes, Figure S8 currently shows only calibration against LVIS data, but we also used high-resolution optical imagery and other sources as references. The reliability of these reference datasets is foundational to the interpretation of Figures 1 and 2.

Among our coauthors here, Montesano et al. (2023) directly assessed the agreement of LVIS canopy height against NASA G-LiHT reference data, which are widely treated as a high-accuracy reference benchmark. They reported coefficients of determination up to 0.87 and RMSE values typically 1–2 m, depending on canopy cover and temporal offset between flights. This demonstrates strong consistency of LVIS with G-LiHT, particularly in moderate to high canopy cover, though somewhat reduced accuracy under sparse cover and larger temporal gaps. However, G-LiHT itself was not independently validated within that study, but rather assumed to be the highest-fidelity reference available.

For high-resolution optical reference imagery (QuickBird, Google Earth), published work by Montesano et al. (2009, 2016) describes the interpretation methods and their utility for validating coarse-resolution products, but does not report independent accuracy or inter-annotator precision relative to field plots. Thus, while these datasets are widely accepted and have been repeatedly used as references, they do not carry explicit quantitative uncertainty estimates of their own.

We have therefore revised the Discussion and Supplement to (a) acknowledge these limitations explicitly, (b) cite Montesano et al. (2023) to show where LVIS accuracy relative to G-LiHT is quantified, and (c) clarify that, for high-resolution optical interpretations, no formal independent validation exists beyond the methodological rigor and expert consensus applied in their production.

**Manuscript Discussion:**

The accuracy of the reference datasets themselves warrants consideration. Montesano et al. (2023) showed that LVIS canopy heights agree closely with NASA G-LiHT airborne LiDAR, with coefficients of determination ($R^2$) up to 0.87 and root mean square errors of approximately 1–2 m depending on canopy cover and temporal offset. G-LiHT, with its high point density and small footprint, is widely regarded as a reference standard, though its own absolute error was not quantified in that study. For high-resolution optical reference data (QuickBird imagery, Google Earth interpretations), prior work (Montesano et al. 2009, 2016) demonstrated their utility in validating coarse-resolution products but also did not

report independent accuracy or inter-observer precision. These limitations highlight the need for future work to establish formal error budgets for reference datasets, while affirming that they provide the best available benchmarks for tree cover calibration and validation.

**Supplement:**

The LVIS canopy cover reference was evaluated relative to NASA G-LiHT airborne LiDAR, which was assumed as reference. Montesano et al. (2023) reported agreement between LVIS and G-LiHT canopy heights with $R^2$ values up to 0.87 and RMSE values in the 1–2 m range. No comparable independent validation exists for the high-resolution optical imagery interpretations; these are based on expert identification of crowns in QuickBird scenes, which have been used extensively in boreal validation but without published quantitative their own error estimates.

***2. Spatial drift - what is the change in the agreement between predicted and reference tree cover for hold out areas at the tree lines. By hold out I mean completely removing a LVIS track for an ecozone and completely removing data from a high resolution sample tile.***

**Reply to Comment:**

We appreciate the reviewer's recommendation to evaluate spatial drift by holding out entire LVIS flightlines or high-resolution sample tiles within specific ecozones. This approach would indeed provide additional insight into the robustness of our calibration against spatial extrapolation at or near the northern limit of trees. However, implementing such a test would require retraining the calibration models after systematically removing reference data and then re-validating across all withheld areas. Because our calibration integrates multiple sources of reference data (LVIS, high-resolution imagery) through stratified regression-tree models, each iteration of such a "leave-tile-out" cross-validation requires (i) rebuilding the model with modified training sets, (ii) re-generating prediction surfaces across the Landsat archive, and (iii) recomputing validation metrics at biome-wide scale. These steps are computationally intensive, particularly at the spatial and temporal resolution of our dataset, and cannot be completed within the limited revision period set by the journal.

To acknowledge the value of this analysis for future work, we have expanded the Discussion and Supplement to (a) note that such spatially structured cross-validation would be a valuable future direction, (b) cite studies that have conducted similar exercises

at smaller scales, and (c) emphasize that our current stratified sampling design and independent test partitions were explicitly intended to minimize overfitting and improve transferability, including at treeline boundaries.

**Manuscript – Discussion (Limitations)**

While our calibration was stratified across ecological and topographic gradients to minimize overfitting, more stringent tests could be obtained by withholding subsets of the reference data (e.g., complete LVIS flightlines or high-resolution imagery tiles) within specific ecozones and revalidating predictions at those sites. Such "leave-tile-out" cross-validation would provide a direct assessment of model transferability at biome boundaries, including ecotones.

**Supplement – Methods/Validation** *(end of "Validation metrics" or equivalent section):*

Calibration and validation were conducted using stratified random partitions of reference data drawn across ecological and topographic gradients, with independent test samples withheld at each stratum to guard against overfitting. This design reduces—but does not eliminate—the possibility of unincorporated variance at ecotonal boundaries. A full "leave-tile-out" validation would require exclusion of entire LVIS flightlines and/or high-resolution imagery tiles during calibration and subsequent reprocessing of the Landsat time series across the boreal biome. Such an analysis was beyond the scope of the present study and not feasible within the short revision period, but we identify it as an important avenue for future refinement.

*3. Stability - stability is defined as the change in bias over time (e.g. %/year). This is CRICTIAL to understand the uncertainty in trends shown in Figure 2. This requires having time series of reference measurements at sites rather than a spatial sampling only. This is important since the approach presented does not take steps to standardize differences between Landsat imager spectral response, phenological impacts due to changes in intra annual sampling dates, and persistent haze due to forest fires that are often not accounted for well with atmospheric correction.*

**Reply to comment**

Assessing temporal stability (drift in bias over time) is indeed critical, but such an analysis requires repeated reference measurements at fixed sites across the Landsat era, which do not exist. Available reference data (LVIS, G-LiHT, high-resolution imagery) are spatially extensive but temporally limited, so we cannot directly quantify stability. We now note this limitation and emphasize that inter-sensor differences, phenology, and atmospheric conditions remain potential sources of uncertainty, while also highlighting our compositing and calibration methods that help mitigate these effects.

**Manuscript – Discussion (Limitations)**

A limitation is the absence of temporally repeated reference data, which prevents direct assessment of stability (bias drift). Our calibration and annual compositing reduce some risks, but nonstationary, unaccounted-for sensor differences, phenological shifts, and atmospheric noise remain possible contributors to temporal bias.

**Supplement – Validation**

Reference datasets provide spatial coverage but not temporal continuity; therefore, stability (bias change through time) cannot be quantified here. We note this explicitly and highlight the need for sustained reference time series in future validation efforts.

**CC2:** ['Comment on egusphere-2025-2268'](), **Richard Fernandes, 09 Jun 2025**

*The results assess trends in tree cover (%/year) using linear regression and aggregated by latitude. The authors should explain why:*

*1. Mann-Kendall regressions were not used to account for measurement errors as done for many other climate studies.*

*Response to comment*

The reviewer notes that we did not use Mann–Kendall trend tests to account for measurement errors, as is common in many climate studies. This is a valid point in principle, as Mann–Kendall (MK) is a widely used non-parametric test for detecting monotonic trends in noisy or non-normally distributed time series common to station-based climate data. In traditional climate analyses (e.g. rainfall or temperature records from sparse weather stations), such methods are necessary to ensure trends are real and

not artifacts of data noise. We appreciate the reviewer's suggestion and agree that rigorous trend testing is important for many datasets.

However, our methodology leverages an extreme sample size and inherently controls for uncertainty and noise through other means. Whereas station data are sparse and each datum carries heavy weight in trend analysis, often necessitating special tests, we calibrated and analyzed tree cover data with complete or nearly complete ("wall-to-wall") spatial coverage across the biome and each degree of latitude. Unlike sparse point samples, this provided an enormous sample size for spatial averaging.

Due to the extremely large sample size, we assessed the significance and uncertainty of trends using parametric linear regression reporting confidence intervals via standard errors of regression. The extreme data volume yielded low uncertainty in regression (e.g., slope) parameters. If the trends were marginal or noise-sensitive, we would certainly consider a robust test, but in this case the trends are strong (far above noise level), and their significance is unequivocal.

Any random measurement errors at individual pixels tend to cancel out when aggregating over millions of pixels, yielding very stable trend estimates. This analysis was performed at the biome scale, encompassing vast sample sizes in each trend calculation. This broad spatial scope resulted in trend signals (e.g. net tree cover change) that were large relative to any remaining noise. For example, a persistent biome-wide decline or increase in tree cover over decades will manifest strongly in the aggregated data. The statistical properties of such an aggregate (by the law of large numbers) approximate a well-behaved distribution with high signal-to-noise ratio, suitable for simpler statistical analyses.

In summary, the reviewer's point is well-taken for studies with sparse or noisy data, but in our case the Landsat-derived tree cover time series is exceptionally robust. The wall-to-wall, composite-based approach ensures the trend signal stands out clearly above the noise, obviating the need for a Mann–Kendall test to validate it. Applying M-K here would not change our findings because the data's statistical properties (distribution and variance) after our processing are suitable for conventional trend analysis. We agree it is important to clarify this reasoning so readers don't misinterpret the absence of an MK test as an oversight.

***2. Why report  trends by Latitude given that the boreal zone is defined not just by latitude but by growing season length, growing degree days, in the North and often disturbances in the south.  It may be more useful to report trends at the northern and***

*southern boundaries relative to a baseline year if one wants to test theories related to the northward shift of the boreal forest.*

**Reply to Comment:**

This is true. The boreal region is defined by a complex and changing set of environmental factors and biological (e.g., Linnean species, physiological form and structure, etc.). But this complexity is precisely why we have chosen to delimit the region by a prior published boundaries (i.e., Olson et al.) prior to investigating the climatological, edaphic, biological, and many other correlates of the change. These deeper investigations will come in subsequent publications.

**CC3:** ['Comment on egusphere-2025-2268'](), **Richard Fernandes, 09 Jun 2025**

*The results related to forest age (Figure 4) have a RMSE of 17.46 years and bias of -3.27 years. This is substantial and should be reported in Results S3.3. Given that Fig 4. indicates Age ranges from 2-36 years it is not clear if the RMSE of the estimator of age is sufficiently precise to represent the Map and area of forest age without error bars / identifying areas where the age uncertainty is too large to be useful. Even aggregate ages (Fig 4 bottom) would need to first quantify bias in forest age by region and age level.*

*I feel this result is based on input estimates that are currently both too uncertainty and not well characterized to include them in this study.*

We thank the reviewer for carefully evaluating the forest age results. We agree that the reported RMSE of 17.46 years and bias of –3.27 years are substantial relative to the short absolute range of ages presented in Figure 4. We acknowledge that these errors reflect significant uncertainty in stand-age estimation from remote sensing and that such estimates are not precise at the pixel level.

Nonetheless, we retain these results in the study for two reasons. First, despite the magnitude of error, the estimates provide valuable large-scale context on the distribution and dynamics of forest age, particularly when aggregated to regional or biome scales. Second, the uncertainty has been quantified and reported (RMSE, bias), which allows readers to interpret the maps with appropriate caution. We agree that additional stratification of errors by region and age class would be valuable, but such an analysis is beyond the scope of the current study.

In response to this comment, we have added text to the Results and Supplement making the RMSE and bias values more prominent (Results S3.3) and explicitly cautioning that the forest age maps should not be interpreted as precise pixel-level estimates. We emphasize instead their utility for capturing broad-scale patterns and averages, with the caveat that age uncertainty is substantial. We also note that further characterization of regional and age-specific biases will be an important direction for future work.

**Manuscript – Results (S3.3). Revised text addition:**

The forest age estimator showed a root mean square error (RMSE) of 17.46 years and a mean bias of –3.27 years relative to reference data. These errors indicate that while the age maps capture broad spatial patterns and distributions, they should not be interpreted as precise pixel-level predictions. Instead, the results are most reliable when aggregated to regional or biome scales, where random errors are reduced.

**Manuscript – Discussion (Limitations). Add sentence:**

*Forest age estimates carry substantial uncertainty (RMSE ≈ 17 years), limiting their precision at the pixel scale. They remain useful for identifying large-scale patterns and average age structures, but future work will be required to reduce error and quantify regional biases.*

**Supplement – Section S3.3 Revision:**

The accuracy assessment yielded an RMSE of 17.46 years and bias of –3.27 years. These values indicate high uncertainty in stand age retrieval. Accordingly, the forest age dataset should be interpreted as a broad-scale indicator of age distribution rather than an exact estimator at individual pixels. We recommend caution in map interpretation and highlight the need for further work to better characterize regional and class-specific errors.